# 18-Item Version of the Short Gambling Harm Screen (SGHS-18): Validation of Screen for Assessing Gambling-Related Harm among Finnish Population

**DOI:** 10.3390/ijerph182111552

**Published:** 2021-11-03

**Authors:** Tiina Latvala, Matthew Browne, Matthew Rockloff, Anne H. Salonen

**Affiliations:** 1Health and Wellbeing Promotion Unit, Department of Public Health Solutions, Finnish Institute for Health and Welfare, 00271 Helsinki, Finland; 2Experimental Gambling Research Laboratory, Central Queensland University, Bundaberg 4670, Australia; m.browne@cqu.edu.au (M.B.); m.rockloff@cqu.edu.au (M.R.); 3Faculty of Health Sciences, University of Eastern Finland, 70211 Kuopio, Finland; anne.salonen@thl.fi

**Keywords:** gambling-related harm, population screen, validation, public health

## Abstract

Background and aims: It is common for gambling research to focus on problem and disordered gambling. Less is known about the prevalence of gambling-related harms among people in the general population. This study aimed to develop and validate the 18-item version of the Short Gambling Harms Screen (SGHS-18). Methods: Population-representative web-based and postal surveys were conducted in the three geographical areas of Finland (*n* = 7186, aged 18 or older). Reliability and internal structure of SGHS-18 was assessed using coefficient omega and via confirmatory factor analysis (CFA). Four measurement models of SGHS-18 were compared: one-factor, six-factor, a second-ordered factor model and a bifactor model (M4). Results: The analysis revealed that only the bifactor model had adequate fit for SGHS-18 (CFI = 0.953, TLI = 0.930, GFI = 0.974, RMSEA = 0.047, SRMR = 0.027). The general factor explained most of the common variance compared to specific factors. Coefficient omega hierarchical value for global gambling harm factor (0.80) was high, which suggested that SGHS-18 assessed the combination of general harm constructs sufficiently. The correlation with the Problem and Pathological Gambling Measures (PPGM) was 0.44, potentially reflecting that gambling harms are closely—although not perfectly—aligned with the mental health issue of problem gambling. SGHS-18 scores were substantially higher for participants who gambled more often, who spent more money or who had gambling problems, demonstrating convergent validity for the screen. Discussion: The SGHS-18 comprehensively measures the domains of gambling harm, while demonstrating desirable properties of internal consistency, and criterion and convergent validity.

## 1. Introduction

### 1.1. Prior Studies

Although gambling is a form of leisure and recreation, gambling participation can be associated with harms [1,2,3]. Although there is not one internationally agreed definition of gambling harm [3,4,5], one practical definition is “any initial or exacerbated adverse consequence due to an engagement with gambling that leads to a decrement to the health or wellbeing of an individual, family unit, community or population” [2]. This definition usefully bases gambling harm within a public-health framework that is distinct from gambling-problems defined as a mental health concern. Based on one framework of harmful gambling [1], any type of gambling can potentially lead to negative consequences. Abbott’s framework highlights eight factors—including cultural, social, psychological, biological, gambling environment, gambling exposure, gambling types, gambling resources—that contribute to gambling harm. Gambling harm in-turn can be defined as an adverse consequence due to engagement with gambling that can lead to a decrement in a person’s health or wellbeing. Instead of focusing on factors that cause harms, Langham’s and colleagues’ [2] alternative framework instead emphasises the outcomes by creating a taxonomy of harms encompassing six domains: financial (e.g., erosion of savings and financial resources), health (e.g., increases in smoking, alcohol consumption, depression) relationship (e.g., harms to relationships, breaches in trust) emotional/psychological (e.g., feelings of shame and stigma), work/study (e.g., reduced performance due to tiredness because of gambling) and social deviance harms. Social deviance harms include: criminal activity, cultural disengagement and neglect of children. Over the past few years, this taxonomy has been used in several studies internationally; including in Australia [6,7], in New Zealand [5], in Canada [8] and in Finland [9,10,11]. 

### 1.2. Prevalence of At-Risk and Problem Gambling

Longer term, much of gambling research has focused on problem and pathological gambling [12,13]. A systematic review including 69 studies showed that there are wide variations in adult past-year problem gambling prevalence rates across different countries varying worldwide from 0.12% to 5.8%, and in Europe from 0.12% to 3.4% [14]. A review including 44 studies on adolescent gambling indicate that problem gambling prevalence rate varied from 0.2% to 12.3%, due to differences among assessment instruments, cut-offs and timeframes [15]. Notably, it is difficult to directly compare studies worldwide [14,16]. Meta-analysis revealed 5 of the 20 available brief (1–5 item) instruments met criteria for satisfactory diagnostic accuracy in detecting both problem and at-risk gambling [13]. Overall, conclusions on reliability and validity of the at-risk and problem gambling instruments were drawn from a relatively limited evidence base. Another review indicated that also evaluation of instruments measuring the reliability at-risk and problem gambling among youth was superficial [17]. 

### 1.3. Prevalence of Gambling-Related Harms

Less is known about the gambling-related harms and their prevalence in the general population [18]. In population-based studies of the impacts of gambling, the most common approach is to use diagnostic screens of pathological or problem gambling [19,20], such as the Problem Gambling Severity Index (PGSI) [21]. However, the proportion of problem gamblers in a population may not provide a full picture of the impacts of harms that can occur in lower risk groups [7]. In particular, although gambling harms are less frequent in low and moderate risk gamblers on a person-by-person basis, they nevertheless may constitute a large proportion of identifiable harms when summed across the whole population, due to the much greater numbers of these gamblers [22]. Prior research on gambling-related harms partially addressed this issue by combining items from the PGSI and the SOGS instruments (The South Oaks Gambling Screen) [23,24], and using this as a proxy to identify harms e.g., [25,26]. However, this is far from an ideal approach, since this was not the original purpose those instruments, which was instead conceptualized to identify problem gambling symptomatology. 

### 1.4. Public Health Perspective

A central principle of the public health perspective is a multifaceted approach to harm reduction in the population, including considerations of product safety, access, and prevention, rather than exclusively focusing on the detection and treatment of problem gamblers [26]. Thus, in order to make the greatest overall reductions in harm, arguably the examination of gambling must shift from narrow addiction-based standpoint to a wider view that covers also low and moderate risk gambling [27]. This highlights the need to study gambling-related harm on the population level rather than exclusively focusing on problem gambling symptomatology. The 72-item Harms Checklist comprehensively measures the negative consequences from gambling. It has been developed using data in Australia and New Zealand [5,7]. However, this measure is not very practical for large-scale population-based studies due to its length.

### 1.5. A Brief Measure of Gambling-Related Harm

Development of a brief measure of public-health impact for gambling specific harm is useful for tracking progress towards harm reduction. Based on the taxonomy of gambling harms [2], the 10-item Short Gambling Harms Screen (SGHS) has been created earlier and validated in an Australian sample [28]. The SGHS was developed based on a selected subset of the 72-item Harms Checklist [2]. While selecting the items, the aim was to maximise sensitivity rather than construct coverage—since the large proportion of harm symptomatology appeared to load satisfactorily on a single main dimension. For the shortest possible measure, it was useful to identify harms that were most prevalent in the population. SGHS has 10 binary-scored items: five from the financial domain, four from the emotion domain and one from relationship domain. Psychometric analysis with the SGHS indicate strong reliability, homogeneity and unidimensionality. However, the screen did not cover all the six domains of gambling-related harms identified in the original checklist. Although this may have been justified in purely statistical terms, the lack of content coverage makes it less attractive as a comprehensive measure of population impact. 

### 1.6. Purpose of This Study

To conclude, creating a screen, which would include the full content coverage of all the domains is desirable as an alternative screen where survey space allows. It has been shown that majority of financial, emotional/psychological and work/study harms are reported by those in the less severe Problem and Pathological Gambling Measure (PPGM) categories. However, among people experiencing problem gambling, health, relationship, and social deviance harms, were the most common harms [29]. In order to help ensure the measure captures impacts to more severely affected individuals, one approach is to capture all of the six domains of gambling-related harms identified in the original 72-item checklist. It has been shown that among persons experiencing more severe gambling problems, we are more likely to observe both more diverse and more severe symptomatology. These new findings support the desirability of measuring these domain-level distinctions in brief measurement instruments. Therefore, the main purpose of this present study was to create and validate a short gambling harm screen that would include all six harm domains. Necessarily, this process involved sacrificing some brevity in the measure to achieve greater coverage of each construct. The goal was to achieve a valid measure of population-level harm that illustrates public health impact, similar to the SGHS, but also reflects items from every domain. 

## 2. Procedure

### Participants

A population-based Gambling Harms Survey was conducted (*n* = 20,000) in the three geographical areas of Finland: Uusimaa, Pirkanmaa and Kymenlaakso [10,11]. The data come from the first wave of a longitudinal population survey. The main purpose of the study was to examine gambling participation, gambling habits, opinions on gambling advertising and experienced gambling-related harm among gamblers and concerned significant others (i.e., persons in a close relationship with someone experiencing gambling problems or gambling-related harm) in the three regions in connection with the reform of the Finnish gambling monopoly by self-report questionnaire. The residents in these three areas cover 42 percentage of the Finnish population. The data were collected by Statistics Finland between January and March in 2017. Participants were randomly selected from the population register. However, 18–24-year-olds were oversampled: they represent 10 percent of the population, but 15 percent of this age group was sampled for the survey. Inclusion criteria included being 18 years old or over, and the ability to understand Finnish or Swedish. Institutionalized persons, such as prisoners, mental health patients and the infirmed were excluded. All participants were invited to the study using a letter, which was sent to their home address retrieved from the national population register system. The invitation letter and the first reminder included a link to the online survey. The next two letters also included the postal questionnaire and a prepaid return envelope. The survey was introduced to the potential respondents as a survey on gambling, gambling-related harm and opinions on gambling marketing. The survey was available in both official languages: Finnish and Swedish. 

After excluding non-eligible individuals (*n* = 67), the study sample size was 19,933 persons. Overall, 7186 adults who were contacted ultimately participated in the study, yielding a response rate of 36.1%. Information about respondents and non-respondents was obtained by combining the study sample with registered based-level socio-demographic data from Statistics Finland. Overall, women and older respondents were more willing to participate than men and younger respondents [10]. 65–74- and 55–64-year-olds were most active respondents while 18–24-year-olds, particularly men in this age group, were least active. In the oldest age group, the response rate was 13 percentage points lower among women than among men. This was the most significant gender difference. Married respondents and those with higher education were more active compared with single or divorced persons or those with lower education. Our use of the term “representative” is intended to convey the population sampling methodology was large-scale and not selective. All population-based studies, such as CATI methodologies and even Census interviews, have some potential for bias from non-responders. Nevertheless, we feel that the description of the study as population-representative is fair based on the reasonably unbiased and large-scale sampling methodology.

Most of the participants answered using the online (71%, *n* = 5084) survey and the rest used the postal survey (29%, *n* = 2102). Gambling, online gambling, at-risk gambling and problem gambling were more common among those who participated using the online survey compared with those using the postal survey [10]. The sample was composed of 47.7% males (*n* = 3426), and the ages of respondents ranged from 18 to 94 years (M = 50.5, SD = 18.8). Respondents who had gambled on at least one game type during the year (2016) were selected from the data for inclusion in the present study (*n* = 5805). More detailed information on demographics and gambling participation of the sample can be seen from Table 1. 

## 3. Measures

### 3.1. Gambling Participation

Gambling participation during the calendar year 2016 was examined by gambling frequency and weekly gambling expenditure. Gambling frequency was asked for 18 pre-defined game types, and an overall gambling frequency was calculated based on the most frequent game type played. Response scale for gambling frequency was daily or almost daily, several times a week, once a week, 2–3 times a month, once a month, less often and not during year 2016. Furthermore, gambling expenditure was probed using the question: ‘Roughly how much money do you spend on gambling (EUR)?’. The respondents were able to select whether to report their expenditure during a typical week, month or year. All results were converted to represent weekly gambling expenditures. Missing expenditure data (4.3%) was not replaced, and all analyses using this data employed listwise deletion.

### 3.2. Problem Gambling

Perceived gambling problem was asked using a question: ‘How often did you think that gambling may have been a problem for you during the year 2016?’. The response options included: Never, sometimes, often, almost always and do not know. A dichotomous variable was created to indicate whether the respondent perceived having gambling problem at least sometimes. Missing data and the option ‘do not know’ were coded as ‘0’ to reflect never.

Problem gambling was examined using the Problem and Pathological Gambling Measure (PPGM) [30]. The PPGM has 14 items which are organized into three sections: Problems (seven questions), Impaired Control (four questions), and Other Issues (three questions). Response scale for all the questions were yes or no. The PPGM was selected since it also included questions on negative consequences of gambling. Furthermore, the PPGM has arguably proven to be the most sensitive and the most accurate instrument in identifying problem gambling [31]. The responses can be categorized: recreational gambling, at-risk gambling, problem gambling and pathological gambling [30]. For the purposes of this study, PPGM was used as continuous variable when examining correlations with other scales and categorial problem gambling variable (two classes: problem gambling and non-problem gambling) where the original problem gambling and pathological gambling classes for the PPGM were combined. Internal scale reliability of the PPGM measured by McDonalds’s omega in the current study was 0.79. 

### 3.3. Gambling-Related Harms

Initially gambling-related harms were evaluated using the 72-item Harms Checklist [2,7]. The item set based on literature review, conceptual framework, and qualitative data [2]. (Langham et al., 2016). To ensure language validity Harms Checklist were translated into Finnish by panel of experts and then back-translated into English in collaboration with the instrument developers. The panel of experts also included a non-participant bilingual collaborator that reversed translations from Finnish back to the original language (English) to verify the quality of the translations. Harms are classified into six domains: financial, health, relationship, emotional/psychological, work/study and social deviance harm. For each of the 72 harm items, a dichotomous variable was created to indicate whether the respondent had experienced such harm in the last 12 months. Respondents were prompted to consider only harms that they perceived were caused by their gambling. The emphasis was to create screen, which would measure subjective experience on gambling harms. McDonalds’s omega value for 72-item Harm Checklist was 0.83.

The Short Gambling Harms Screen (SGHS) was calculated from a select subset of the 72-item Harms Checklist [30]. It has 10 binary scored items, five from the financial domain (reduction of my available spending money, reduction of my savings, less spending on recreational expenses such as eating out, going to movies or other entertainment, sold personal items, increased credit card debt) four from the emotions domain (had regrets that made me feel sorry about my gambling, felt ashamed of my gambling, felt distressed about my gambling, felt like a failure) and one from the relationship domain (spent less time with people I care about). Psychometric analysis with the SGHS indicate very strong reliability, homogeneity and unidimensionality [7]. In this study, McDonalds’s omega of the SGHS was also moderately high (0.68).

## 4. Statistical Analysis

From each six domains of the 72-item Harms Checklist, three items were selected for the new 18-item version (SGHS-18). The item selection process was based on Browne et al.’s [28] approach. The general principle is to simultaneously minimise false negatives when predicting the presence of harm in the complete screen, whilst also selecting items that capture the entire construct of gambling harm. Within each domain, the first item was selected based the highest prevalence. The second item for each domain was chosen based on the maximum prevalence amongst cases who have not answered positively on the previously selected item within that same domain. This algorithm tends to maximise sensitivity, whilst also ensuring that diverse and non-redundant items were selected, so as to cover the entire construct within each domain. The selected 18 items are presented in Table 2.

First, exploratory factor analysis was conducted for the original 72-item checklist and a parallel analysis of the scree plot was used to discover overall factor structure. Parallel analysis of the scree plot showed that one or six factor structure would fit the data best (results not shown). Then confirmatory factor analysis for SGHS-18 was used to estimate the degree of fit of four measurement models to the data. The starting point was a first-order one-factor model which is called M1. The next model (M2) included six first-order factors. In the third model (M3) in addition to the six first-order factors one second-order factor was specified representing the global harm dimension and the covariances between factors. The fourth model (M4) included a bifactor model representing a global harm dimension on which each item is loaded and six harm specific factors where correlations between factors were fixed to zero. These results are presented in Table 3.

Measurement invariance implicates that the same construct is being measured across some specified groups, like sex and age. In CFA invariance can be tested by comparing models with parameters constrained and unconstrained between groups. However, due to high amount of 0 values, we were not able to examine measurement invariance with respect to important demographics.

In CFA diagonally weighted least squares (DWLS) estimation was used. Models were compared by statistical tests. These were comparative fit index (CFI), root mean square error of approximation (RMSEA), the goodness of fit index (GFI), Tucker Lewis index (TLI) and the standardised root mean square residual (SRMR). The cut-off criteria for good fit for GFI ≥ 0.90–0.95, for CFI ≥ 0.90, for RMSEA ≤ 0.08, for TLI ≥ 0.95 and for SRMR ≤ 0.08 [32]. (Hooper, Coughlan, and Mullen, 2008). Bifactor model was judged by the omega hierarchical indices, which measure how precisely screen score assesses the combination of general and specific constructs [33]. Additionally, an explained common variance index (ECV) was used, in order to quantify the degree of unidimensionality in bifactor model [34]. This is presented in Table 4.

To investigate convergent validity, correlations between the summed 72-item Harms Checklist, the SGHS, the PPGM and the SGHS-18 were calculated (Table 5). For examining concurrent validity, one-way analysis of variance (one-way ANOVA) was conducted to examine whether SGHS-18 scores for those who gambled more frequently or spend more money on gambling, would be higher than for those who gambled less. Likewise, SGHS-18 scores were compared by one-way ANOVA between those who considered as having gambling problems and those who do not considered (Table 5). All the analyses were run in R (version 3.2.1, R Core Team, Vienna, Austria). 

### Ethics

The Ethics Committee of the Finnish Institute for Health and Welfare approved the research protocol. Potential participants received written information about the study and the principles of voluntary participation. The basic principles of the research ethics were followed throughout the research process (The World Medical Association’s Declaration Helsinki 2004).

## 5. Results

### Reliability and Internal Structure of the SGHS-18

The selected 18 items and the percentage of positive response (PR) to these items are presented in Table 2. Table 2 also contains the progressive number of false negatives, which indicates proportion of non-zero responses on the current subset, relative to the non-zero responses on the 72-item checklist (FN). The running Spearman correlation of the subset sum with the full harms sum is also given (STC) (for example correlation between sum of first item “Reduction of my available spending money” and full harms sum, then correlation between sum of first and second items “Reduction of my available spending money” and “Less spending on recreational expenses such as eating out going to movies or other entertainment”, and full harms sum). One can see that saturation with respect to both construct coverage/subset-total correlation (0.98), and percentage of false negatives (0.5%) is achieved at around 18 items.

Table 3 shows the results of all the CFA models tested. Based on guidelines presented earlier only the bifactor model had acceptable fit (CFI= 0.953, TLI = 0.930, GFI = 0.974, RMSEA = 0.047, SRMR = 0.027). Table 4 presents the standardized factor loadings for the bifactor model. All items loaded significantly on global harm factor. With the exception of one item, (Spent less time with people I care about) all the items loaded also significantly on their specific factors. This item seemed to represent only global harm rather than the specific harm symptoms. 

Table 4 also includes the ECV and omega hierarchical of the general and specific factors. ECV for the general factor was 0.60, and the ECV-values for the specific factors were 0.09, 0.07, 0.06, 0.09, 0.05 and 0.03, respectively. The omega was 0.80, and the values for specific factors were much lower (0.30, 0.40, 0.25, 0.36, 0.00, 0.02). These findings imply that only the general factor has sufficient variance and reliability for meaningful interpretation. Due to this validity, examinations are conducted only for SGHS-18.

## 6. Validity

SGHS-18 was strongly correlated (0.98) with the 72-item Harms checklist, which indicated that the SGHS-18 captured the primary construct of gambling-related harm (Table 5). 

Likewise, SGHS-18 correlated strongly with the original SGHS (0.97). However, the correlation with the PPGM was weaker (0.44) but still moderate, which was expected given that the PPGM is a measure of problem gambling, and not exclusively gambling-harm. 

Participants who had SGHS-18 score greater than zero (12%) had an average PPGM score of 1.7, compared to 0.12 for those scoring zero on SGHS-18 (t = 23.74, *p* ≤ *0*.001, Cohen’s d = 0.975). When SGHS-18 scores were examined by gambling frequency, gambling expenditure and by perceived gambling problem, SGHS-18 scores were higher for participants who gambled more often, who spent more money or who had gambling problems. This demonstrated the property of convergent validity. For example, those who were experiencing gambling problems had an average score of 2.36 on the SGHS-18 compared to 0.11 for those who did not consider themselves having a problem (see Table 6).

## 7. Discussion

Using the item pool of 72 specific harms caused by gambling [2,7], the purpose of this study was to develop a screen for gambling-related harm that covers all identified domains of harm outlined by Langham et al. [2]. The emphasis was to include items from all six domains of the original 72 item Harms Checklist to cover the entire spectrum of gambling from recreational gambling to problem gambling. Confirmatory factor analysis was used to estimate the degree of fit of four measurement models to the data. Only the bifactor model had acceptable fit. ECV and omega hierarchical for the general factor were much higher than for specific factors. Based on these results, the SGHS-18 appeared to be unidimensional. Over half (60%) of the variance was shared with the general factor and thus it accounted for around 1.5 times more common variance than the six specific factors together. 

The SGHS-18 was well correlated with both the 72-item checklist and the 10-item SGHS. Correlation with PPGM total (0.44) was only moderate. Importantly, however, the PPGM is primarily a measure of problem gambling severity. “Problem gamblers” at least sometimes during the calendar year 2016, had substantially higher SGHS-18 scores. Higher scores were also evident for participants who gambled more frequently and spent more money on gambling. This is in line with earlier studies, and with the definition of problem gambling; wherein harm from gambling is understood to derive from frequent and excessive investment of time and money [2,35]. 

Negative consequences of gambling occur also among those who do not meet the criteria of potential problem gambling [7]. Therefore, gambling-related harms appear to be not solely a consequence of a mental health condition, but also result from people simply engaging with gambling at excessive levels. However, many studies concentrate on clinical screens and accordingly assess only fraction of the population that is experiencing harm. From a prevention standpoint, it is important to concentrate broadly on those experience negative harms rather than only those who are problem gamblers. SGHS-18 is an alternative longer, but more comprehensive population screen than the SGHS; covering all domains, especially those which are more commonly present among problem gamblers. This should lead to greater sensitive across the entire spectrum of gamblers. There is, however, only a minor sacrifice in terms of brevity and on time required from respondents. The SGHS-18 is quick to administer, measures commonly reported harms and has strong psychometric properties. The screen was associated with gambling frequency, gambling expenditure and perceived gambling problems. SGHS-18 can be used in contexts when the aim is to achieve sensitive and valid monitoring of the population-level impact of gambling. 

There were some limitations in the study. The response rate of this study (36%) was better than the international average for web-based and postal problem gambling surveys [16]. Gambling-related harm was probed based on framework that measures the harms comprehensively [2,7]. Furthermore, both the Harms Checklist and the PPGM were translated into Finnish and back-translated into English in collaboration with the instrument developers. However, so far, the psychometric properties of the PPGM have not been studied in the Finnish context, and due to survey-space limitations we did not use any other validated scales of problem gambling apart from the PPGM. The method of calculating overall gambling frequency might not reflect on the actual overall gambling frequency accurately for those who use play multiple types of games. Calculation based on the most frequently used game type might underestimates the actual gambling frequency for people who play multiple games, or equivalently underestimate frequency for people who play few or one gambling game(s). For further studies it would be useful to examine how SGHS-18 would correlate with some wellbeing scales. For example, quality of life, as Browne and colleagues [6] did in Australia. Similar to the original SGHS, the SGHS-18 provides only a score indicator of degree of harm. Future work might consider linking these scores to recognised population health metrics of health, wellbeing and morbidity. As our sample had many participants who did not experience any gambling harms (there were many 0 values), we were not able to examine measurement invariance with respect to important demographics (e.g., age, gender, etc.). It is conceivable that different demographic groups may experience different types of harm [10]. Accordingly, future work might study measurement invariance of the SGHS-18 between with respect to gender, age, and other salient categories. Further, low prevalence rates on the items of gambling harm might have influenced the findings of the study due to leverage on these items, where only a few participants contributed to the significance of results related to single-item indicators of harm. Moreover, the cross-sectional design was not conducive to exploring causal relationships between gambling harm and the presumed antecedent variables. Further, it must be recognised that in absence of probing all 72 harms, respondents may experience other harms that were not detected by any short instrument (see [26]). In addition, gambling-related harm influence not only gamblers themselves, but also their families, friends, work places and the whole community [1,2,21,36]. 

## 8. Conclusions

To conclude, reliability and internal structure of SGHS-18 was examined by coefficient omega (hierarchical) and using confirmatory factor analysis (CFA). Hierarchical coefficient omega value of the SGHS-18 was high, which suggested very good classical reliability. Additionally, a bifactor CFA showed good properties, although ECV and omega hierarchical values for the general factor were much higher compared to specific factors, which suggested unidimensional structure of SGHS-18. The correlation with the Problem and Pathological Gambling Measures (PPGM) was weaker, which supports the idea that harm and pathology are related, but distinct constructs: that they may occur independently. When SGHS-18 scores were examined by gambling frequency, gambling expenditure and by perceived gambling problem, harm scores were higher for participants who gambled more often, who spent more money or who had gambling problems. Therefore, the 18-item version of the Short Gambling Harms Screen (SGHS-18) measures harm domains comprehensively, and furthermore demonstrated good fit to the data and very good internal reliability. 

The SGHS-18 can be used to monitor changes in gambling among population. It may be more sensitive to capture minor changes compared to problem gambling measures. This measure ought to allow for better quantification of the degree of gambling harm experienced across the spectrum of severity. This allows us to shift our interest from simply identifying the number of problem gamblers within a population to prouder public health perspective that focus on morbidity: the degree to which the problems are affecting quality of life. However, we suggest that measuring harms should be extended beyond the gambler also to their friends, family, community and society.

## Figures and Tables

**Table 1 ijerph-18-11552-t001:** Respondents’ gender, age, gambling frequency, weekly gambling expenditure and perceived gambling problem during the year 2016.

	% (*n*)
Gender	
Male	50.1 (2954)
Female	49.9 (2937)
Age	
18–34	26.6 (1564)
35–54	34.9 (2053)
55–74	30.8 (1815)
≥75	7.8 (458)
Gambling frequency	
Daily or almost daily	4.1 (241)
Several times a week	7.1 (415)
At least once a week	29.1 (1689)
2–3 times a month	13.4 (780)
Once a month	11.5 (670)
Less often	34.0 (1971)
Weekly gambling expenditure	
Less than EUR 5	65.4 (3386)
EUR 5–10	15.6 (807)
EUR 11–20	10.4 (539)
EUR 21 or over	8.6 (448)
Perceived gambling problem	6.4 (371)
Problem gambling ^1^	2.4 (139)
Experienced gambling-related harm ^2^	12.5 (725)

^1^ PPGM, Problem and Pathological Gambling Measure; ^2^ Harms Checklist.

**Table 2 ijerph-18-11552-t002:** Properties of the 18 selected harm items with respect to the full checklist.

Category	Item	PR (%)	FN (%)	STC (r)	ITC (r)
Financial	Reduction of my available spending money	6.3	6.23	0.694	0.694
Financial	Less spending on recreational expenses such as eating out, going to movies or other entertainment	1.9	5.39	0.742	0.394
Financial	Reduction of my savings	2.9	4.32	0.800	0.486
Work/Study	Used my work or study time to gamble	0.5	3.82	0.801	0.216
Work/Study	Reduced performance at work or study	1.0	3.77	0.825	0.281
Work/Study	Used my work or study resources to gamble	0.2	3.72	0.828	0.130
Health	Loss of sleep due to spending time gambling	0.8	3.58	0.831	0.267
Health	Increased my use of tobacco	0.7	3.53	0.838	0.246
Health	Increased experience of depression	0.8	3.53	0.841	0.261
Emotional/Psychological	Had regrets that made me feel sorry about my gambling	1.4	3.20	0.858	0.346
Emotional/Psychological	Felt like a failure	5.0	1.26	0.944	0.627
Emotional/Psychological	Felt ashamed of my gambling	2.1	0.76	0.966	0.411
Relationships	Spent less time with people I care about	0.8	0.64	0.971	0.264
Relationships	Spent less time attending social events	0.5	0.62	0.972	0.218
Relationships	Experienced greater tension in my relationships	0.4	0.55	0.976	0.184
Social deviance	Reduced my contribution to community obligations	0.3	0.52	0.977	0.160
Social deviance	Outcast from community due to involvement with gambling	0.3	0.52	0.977	0.171
Social deviance	Promised to pay back money without genuinely intending to do so	0.3	0.52	0.977	0.151

PR percent positive responses, FN false negatives (incremental), STC subscale to 72-item total correlation (Spearman), ITC item 72-item total correlation.

**Table 3 ijerph-18-11552-t003:** One-factor (M1), six-factor (M2), a second-ordered factor model (M3) and a bifactor model (M4).

	Model	CFI	TLI	RMSEA	SRMR	GFI
1-factor first-order model	M1	0.830	0.807	0.078	0.054	0.906
6-factor first-order model	M2	0.911	0.866	0.060	0.038	0.953
Second-order factor model with six first-order factors	M3	0.870	0.846	0.069	0.048	0.932
Bifactor model with general factor and six first-order factors in an orthogonal structure	M4	0.953	0.930	0.047	0.027	0.974

Comparative fit index (CFI), root mean square error of approximation (RMSEA), the goodness of fit index (GFI), Tucker Lewis index (TLI) and the standardised root mean square residual (SRMR).

**Table 4 ijerph-18-11552-t004:** Standardized factor loadings of the bifactor model of SGHS-18.

Item	FIN	WORK	HEL	EMO	REL	SOC	Global
Reduction of my available spending money	0.52						0.27
Reduction of my savings	0.40						0.37
Less spending on recreational expenses such as eating out, going to movies or other entertainment	0.48						0.38
Reduced performance at work or study		0.54					0.51
Used my work or study time to gamble		0.44					0.27
Used my work or study resources to gamble		0.18					0.23
Loss of sleep due to spending time gambling			0.40				0.59
Increased my use of tobacco			0.44				0.41
Increased experience of depression			0.34				0.62
Felt ashamed of my gambling				0.43			0.45
Had regrets that made me feel sorry about my gambling				0.56			0.29
Felt like a failure				0.37			0.44
Spent less time with people I care about					0.01		0.61
Spent less time attending social events					0.44		0.81
Experienced greater tension in my relationships					−0.44		0.56
Outcast from community due to involvement with gambling						0.30	0.55
Reduced my contribution to community obligations						−0.24	0.60
Promised to pay back money without genuinely intending to do so						0.28	0.49
ECV	0.09	0.07	0.06	0.09	0.05	0.03	0.60
Omega	0.60	0.53	0.71	0.63	0.75	0.60	0.88
Omega hierarchical	0.40	0.30	0.25	0.36	0.00	0.02	0.80

**Table 5 ijerph-18-11552-t005:** Spearman correlations of the 18-item version of the Short Gambling Harm (SGHS-18) with Harms checklist, the Short Gambling Harm Screen (SGHS) and the Problem and Pathological Gambling Measure (PPGM) as a continuous variable.

	SGHS-18	SGHS (10 Items)	Harms Checklist (72 Items)
SGHS-18	-		
SGHS (10 items)	0.97	-	
Harms Checklist (72 items)	0.98	0.95	-
PPGM	0.44	0.43	0.45

**Table 6 ijerph-18-11552-t006:** Average SGHS-18 scores examined by gambling frequency, gambling expenditure and by perceived gambling problems.

	(M)	Test Statistic	Cohen’s F	Post Hoc
Gambling frequency	SGHS-18	F(5,5760) = 101.2 ***	0.30	
1. Daily or almost daily	1.51			1 > 2–6
2. Several times a week	0.78			2 > 3–6
3.Weekly	0.21			3 > 6
4. 2–3 times a month	0.19			ns.
5. Once a month	0.16			ns.
6. Less than monthly	0.11			ns.
Gambling expenditure			0.35	
1. EUR < 5	0.12	F(3,5176) = 214.7 ***		1 < 2–4
2. EUR 5–10	0.26			2 < 4
3. EUR 11–20	0.33			3 < 4
4. EUR > 20	1.46			4 > 1–3
Perceived gambling problem			0.60	
Yes	2.36	F(1,5623) = 1994.7 ***		
No	0.11			

*** *p* < 0.001.

## Data Availability

The survey data is publicly accessible for research purposes from the Finnish Society Science Data Archive (FSD) with the name of Rahapelikysely 2016 (FSD3261), urn:nbn:fi:fsd:T-FSD3261.

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
