# Peer review of "18-Item Version of the Short Gambling Harm Screen (SGHS-18): Validation of Screen for Assessing Gambling-Related Harm among Finnish Population"

_ijerph, 2021, doi:10.3390/ijerph182111552_

Round 1

Reviewer 1 Report

Your work has touched upon the issue of gambling, one of the most important problems of our time. I want to congratulate you on researching such an important social issue.
Sampling is quite a lot and sufficient in your work. The SEM values you use are within an acceptable cut-off, the literature is adequate, the results and discussion are quite understandable. 

However, I have some concerns about the paper, and I consider that there are several limitations that prevent me from recommending the publication of the study in its current form. Following are my major comments and questions:

  1. While using the term "screen" in the article's title, "scale" is used in some places in the text. How would you fix it as "scale" in the title?
  2. It could be relevant for adding meaning to the manuscript and awaking the interest of the readers, to add a paragraph in the introduction section in which the authors explain more about the necessity of conducting this research. There is a lack of studies in this topic? Why is important your paper? How do you contribute to the lack of literature on this topic? What are the contributions you what to do? What is the problem that you what to solve?
  3. There is not enough information about how you ensure language validity in your study. In other words, you should explain in detail the languages of the original scale and the shortened scale, and the language validity if there is a difference.
  4. Since there is a language change in the scale, it must first be EFA, then CFA. Why did you perform CFA without applying EFA in factor analysis? Especially in the adaptation study, weren't you supposed to do EFA first?
  5. Why didn't you explain the details that measurement invariance or measurement equivalence?
  6. Why the hypothesis and research questions were not done for validity tests.
  7. You gave the Cronbach Alpha value on the scales, but you mentioned Omega above. My advice to you is to consider the McDonald' Omega (1999) instead of Cronbach's alpha. When using alpha for Likert-type rating responses scales the magnitude of coefficient alpha can be deflated if loadings are not equal across all items (Yang & Green, 2010) and if the nature of the data is not continuous (Elosua & Zumbo, 2008). So that, it is recommended the use of McDonalds's Omega (1999) because it has shown evidence of better accuracy and factor loadings do not need to be equal for all items (Zhang & Yuan, 2016).
  8. It could be specified that it is a self-report questionnaire in method section.
  9. This study should have had limitations. Before conclusions, limitations can be written in terms of scales, method, data collected, timing and etc.
  10. In discussion and conclusion, I encourage the authors to better highlight the new knowledge gained by this study As well as some implications for future research, I think there could be more. The conclusions could be deepened pls."

Author Response

We are very grateful for the time and detailed comments provided by reviewers.  We have incorporated the suggested changes into the manuscript by “Track Changes” function.

Reviewer 1:

1. While using the term "screen" in the article's title, "scale" is used in some places in the text. How would you fix it as "scale" in the title?

Response: Term scale is replaced with screen in the text.

2. It could be relevant for adding meaning to the manuscript and awaking the interest of the readers, to add a paragraph in the introduction section in which the authors explain more about the necessity of conducting this research. There is a lack of studies in this topic? Why is important your paper? How do you contribute to the lack of literature on this topic? What are the contributions you what to do? What is the problem that you what to solve?

Response: As it is mentioned in the introduction development of a brief measure of public-health impact for gambling specific harm is useful for tracking progress towards harm reduction. 10-item Short Gambling Harms Screen (SGHS) has been created earlier and validated in an Australian sample. However, it covers only financial, emotion and relationship domain. It has been shown that majority of financial, emotional/psychological, and work/study harms are reported by those in the less severe Problem and Pathological Gambling Measure (PPGM) categories. However, among people experiencing problem gambling, health, relationship, and social deviance harms, were the most common harms (Browne et. al. 2020). In order to help ensure the measure captures impacts to more severely affected individuals, one approach is to capture all of the six domains of gambling-related harms identified in the original 72-item checklist. It has been shown that among persons experiencing more severe gambling problems, we are more likely to observe both more diverse and more severe symptomatology Text bolded is added to page 3.

3. There is not enough information about how you ensure language validity in your study. In other words, you should explain in detail the languages of the original scale and the shortened scale, and the language validity if there is a difference.

Response: To ensure language validity Harms Checklist were translated into Finnish by panel of experts and then back-translated into English in collaboration with the instrument developers. Panel of expertise also included non-participant bilingual collaborator that reversed translations from Finnish back to the original language (English) to verify the quality of the translations. This is added to page 5.

4. Why didn't you explain the details that measurement invariance or measurement equivalence?

Response: This is added to page 6: Measurement invariance implicates that the same construct is being measured across some specified groups, like sex and age. In CFA invariance can be tested by comparing models with parameters constrained and unconstrained between groups. However, due to high amount of 0 values, we were not able to examine measurement invariance with respect to important demographics.

5. Since there is a language change in the scale, it must first be EFA, then CFA. Why did you perform CFA without applying EFA in factor analysis? Especially in the adaptation study, weren't you supposed to do EFA first?

Response: EFA was conducted for the original 72-item checklist (results not shown) and a parallel analysis of the scree plot showed that one or six factor structure would fit the data best. This is added to page 6: First exploratory factor analysis was conducted for the original 72-item checklist and a parallel analysis of the scree plot was used to discover overall factor structure. Parallel analysis of the scree plot showed that one or six factor structure would fit the data best (results not shown). Then confirmatory factor analysis for SGHS-18 was used to estimate the degree of fit of four measurement models to the data.

6. Why the hypothesis and research questions were not done for validity tests.

Response: We are sorry, but we did not understand the question.

7. You gave the Cronbach Alpha value on the scales, but you mentioned Omega above. My advice to you is to consider the McDonald' Omega (1999) instead of Cronbach's alpha. When using alpha for Likert-type rating responses scales the magnitude of coefficient alpha can be deflated if loadings are not equal across all items (Yang & Green, 2010) and if the nature of the data is not continuous (Elosua & Zumbo, 2008). So that, it is recommended the use of McDonalds's Omega (1999) because it has shown evidence of better accuracy and factor loadings do not need to be equal for all items (Zhang & Yuan, 2016).

Response: Omega values are presented instead, as suggested.

8. It could be specified that it is a self-report questionnaire self-report questionnaire in method section.

Response: This is specified and added to page 3.

9. This study should have had limitations. Before conclusions, limitations can be written in terms of scales, method, data collected, timing and etc.

Response: There are seven limitations already mentioned in text, presented also above: (it is summarized on page 9 that there were limitations to the study)

  • relatively low response rate,
  • the method of calculating overall gambling frequency might not reflect on the actual overall gambling frequency accurately for those who use play multiple types of games. Calculation based on the most frequently used game type might underestimates the actual gambling frequency for people who play multiple games, or equivalently under-estimate frequency for people who play few or one gambling game(s).
  • the psychometric properties of the PPGM have not been studied in the Finnish context, and due to survey-space limitations we did not use any other validated scales of problem gambling apart from the PPGM,
  • because our sample had many participants who did not experience any gambling harms (there were many 0 values), we were not able to examine measurement invariance with respect to important demographics (e.g., age, gender, etc.). It is conceivable that different demographic groups may experience different types of harm (Salonen, Latvala, et al., 2017)
  • low prevalence rates on the items of gambling harm might have influenced the findings of the study due to leverage on these items, where only a few participants contributed to the significance of results related to single-item indicators of harm
  • the cross-sectional design was not conducive to exploring causal relationships between gambling harm and the presumed antecedent variables
  • absence of probing all 72 harms, respondents may experience other harms that were not detected by any short instrument

10. In discussion and conclusion, I encourage the authors to better highlight the new knowledge gained by this study As well as some implications for future research, I think there could be more. The conclusions could be deepened pls."

Response: SGHS-18 is an alternative longer, but more comprehensive population scale than the SGHS; covering all domains, especially those which are more commonly present among problem gamblers. Thus, we can reach the whole spectrum of gamblers. There is, however, only a minor sacrifice in terms of brevity and on time that it takes from respondent. Bolded texts are added to page 9.

SGHS-18 can be used to monitor changes in gambling among population. It may be more sensitive to capture minor changes compared to for example problem gambling measures. Overall, this study can allow us better understand gambling harms and increase their visibility in population by making harms measurable. This allows shifting our interest from simply identifying the number of problem gamblers within a population to broader public health perspective. Further, this can help us better understand how harms can be reduced and target more effectively through interventions. However, we suggest that measuring harms should be extend beyond the gambler to their friends, family, community and society. Bolded texts are added to page 11.

Reviewer 2 Report

Manuscript No.: ijerph-1404199

Manuscript Title: 18-item version of the Short Gambling Harm Scale (SGHS-18): Validation of a scale for assessing gambling-related harm among Finnish population

This article aims to develop and validate the 18-item version of the Short Gambling Harms Screen (SGHS-18). One of the key contributions that the authors claimed was offering essential implications for the gambling-related harms and their prevalence in the general population in the gambling context. Accordingly, a confirmatory factor analysis approach was adopted. I have several major concerns about the quality and contribution of the study, as detailed below.

  1. The intended contribution of the paper seems to rest on authors’ claim on this study trying to achieve a valid measure of population-level harm that illustrates public health impact, similar to the SGHS, but also reflects items from every domain. However, after revealing the results, I am not sure how this study could address the inconsistency. I also question the validity of the coding process (e.g., perceived gambling problem), which seems to be rather subjective.

  1. As the novel coronavirus has spread across the world, collateral damage has been far and wide, with many industries sustaining one of the heaviest blows. Readers may want to see the COVID-19’s devastating effect on consumers’ perception regarding different gambling behaviors, attitudes, etc. in the introduction section.

  1. The literature review section is missing and should deserve more attention.

  1. The methodology should be revised to better justify the operational definition of the variables of interest. Given that data were driven from data collected from “A population-based Gambling Harms Survey ”, the authors are suggested to take a qualitative approach in identifying the codes with validity intact.

======================================================

Hope the above comments helpful. Thank you again.

Author Response

Reviewer 2:

We are very grateful for the time and detailed comments provided by reviewers.  We have incorporated the suggested changes into the manuscript by “Track Changes” function.

1. The intended contribution of the paper seems to rest on authors’ claim on this study trying to achieve a valid measure of population-level harm that illustrates public health impact, similar to the SGHS, but also reflects items from every domain. However, after revealing the results, I am not sure how this study could address the inconsistency. I also question the validity of the coding process (e.g., perceived gambling problem), which seems to be rather subjective.

Response: As it is mentioned in the introduction development of a brief measure of public-health impact for gambling specific harm is useful for tracking progress towards harm reduction. 10-item Short Gambling Harms Screen (SGHS) has been created earlier and validated in an Australian sample. However, it covers only financial, emotion and relationship domain. It has been shown that majority of financial, emotional/psychological, and work/study harms are reported by those in the less severe Problem and Pathological Gambling Measure (PPGM) categories. However, among people experiencing problem gambling, health, relationship, and social deviance harms, were the most common harms (Browne et. al. 2020). In order to help ensure the measure captures impacts to more severely affected individuals, one approach is to capture all of the six domains of gambling-related harms identified in the original 72-item checklist. It has been shown that among persons experiencing more severe gambling problems, we are more likely to observe both more diverse and more severe symptomatology Text bolded is added to page 3.

All items were based on Langham’s et al. (2015) work and developed based on literature review, focus groups and interviews with professionals involved in support and treatment services for gambling problems, interviews with people who gamble and their affected others, and the analysis of public forum posts of people experiencing problems with gambling and their affected others. Consequently, these harms are face-valid with respect to being how gamblers describe their own experiences of harm. We have not tried to impose our values upon these descriptions since any critique of the items is implicitly a critique of these gambler’s experiences as they have expressed them. Respondents were prompted to consider only harms that they perceived were caused by their gambling. To conclude, the emphasis was not to create measure, which would be objective, on the contrary we were interested on subjective experience on gambling harms.

This is added to page 5: The item set based on literature review, conceptual framework, and qualitative data. This to page 6: Respondents were prompted to consider only harms that they perceived were caused by their gambling. The emphasis was to create screen, which would measure subjective experience on gambling harms.

2. As the novel coronavirus has spread across the world, collateral damage has been far and wide, with many industries sustaining one of the heaviest blows. Readers may want to see the COVID-19’s devastating effect on consumers’ perception regarding different gambling behaviors, attitudes, etc. in the introduction section.

Response: Although this is interesting and timely subject, this topic is beyond our interest. In addition, given the nature data collected, there was limited scope to explore this question.

3. The literature review section is missing and should deserve more attention.

Response: This is added to review section on page 2:  A systematic review including 69 studies showed that there are wide variations in adult past-year problem gambling prevalence rates across different countries varying worldwide from 0.12% to 5.8%, and in Europe from 0.12% to 3.4% (Calado & Griffiths 2016). A review including 44 studies on adolescent gambling indicate that problem gambling prevalence rate varied from 0.2% to 12.3 %, due to differences among assessment instruments, cut-offs, and timeframes (Calado et al. 2016). Notably, it is difficult to directly compare studies worldwide (Williams et al. 2012; Calado & Griffiths 2016). Meta-analysis revealed five of the 20 available brief (1-5 item) instruments met criteria for satisfactory diagnostic accuracy in detecting both problem and at-risk gambling (Dowling et al. 2019). Overall, conclusions on reliability and validity of the at-risk and problem gambling instruments were drawn from a relatively limited evidence base. Another review indicated that also evaluation of instruments measuring the reliability at-risk and problem gambling among youth was superficial (Edgren et al. 2016). 

4. The methodology should be revised to better justify the operational definition of the variables of interest. Given that data were driven from data collected from “A population-based Gambling Harms Survey ”, the authors are suggested to take a qualitative approach in identifying the codes with validity intact.

Response: All items were based on Langham’s et al. (2015) work and developed based on literature review, focus groups and interviews with professionals involved in support and treatment services for gambling problems, interviews with people who gamble and their affected others, and the analysis of public forum posts of people experiencing problems with gambling and their affected others. Consequently, these harms are face-valid with respect to being how gamblers describe their own experiences of harm. We have not tried to impose our values upon these descriptions since any critique of the items is implicitly a critique of these gambler’s experiences as they have expressed them.

Round 2

Reviewer 2 Report

Manuscript No.: ijerph-1404199

Manuscript Title: 18-item version of the Short Gambling Harm Screen (SGHS-18): Validation of a screen for assessing gambling-related harm among Finnish population

Thank you for your work in revising your paper. Editors feel it sets out an important and interesting issue, and is a valuable addition to our previously published articles on too much measurement instruments. I would like to congratulate the authors for a job well done on a topic which badly needs addressing. Overall, this revised revision has better balance. Some of the remaining possibly better reflects my perspective on the topic and the authors may wish to present alternative perspectives to this review. There are still several points that need authors pay attention.

The authors have done an admirable job summarizing the relevant literature and outlining the rationale as to why we need to question certain gambling-related harms and their prevalence in the general population. A literature review is part of the introduction, but most often, a literature review is formatted to appear as a separate section of your paper, preceding the body. So, I suggest authors could make it a separate section.

======================================================

Hope the above comments helpful. Thank you again.

Author Response

We would like to thank Reviewer 2 on her/his time and  comments. We tried to make introduction  more fluent and easier to follow by making it as separate section as suggested  and adding subheadings to it.